# MEASURING COMPOSITIONALITY IN REPRESENTATION LEARNING

**Jacob Andreas**
Computer Science Division
University of California, Berkeley
`jda@cs.berkeley.edu`

## ABSTRACT

Many machine learning algorithms represent input data with vector embeddings or discrete codes. When inputs exhibit compositional structure (e.g. objects built from parts or procedures from subroutines), it is natural to ask whether this compositional structure is reflected in the the inputs' learned representations. While the assessment of compositionality in languages has received significant attention in linguistics and adjacent fields, the machine learning literature lacks general-purpose tools for producing graded measurements of compositional structure in more general (e.g. vector-valued) representation spaces. We describe a procedure for evaluating compositionality by measuring how well the true representation-producing model can be approximated by a model that explicitly composes a collection of inferred representational primitives. We use the procedure to provide formal and empirical characterizations of compositional structure in a variety of settings, exploring the relationship between compositionality and learning dynamics, human judgments, representational similarity, and generalization.

## 1 INTRODUCTION

The success of modern representation learning techniques has been accompanied by an interest in understanding the structure of learned representations. One feature shared by many *human-*designed representation systems is compositionality: the capacity to represent complex concepts (from objects to procedures to beliefs) by combining simple parts (Fodor & Lepore, 2002). While many machine learning approaches make use of human-designed compositional analyses for representation and prediction (Socher et al., 2013; Dong & Lapata, 2016), it is also natural to ask whether (and how) compositionality arises in learning problems where compositional structure has not been built in from the start. Consider the example in Figure 1, which shows a hypothetical character-based encoding scheme learned for a simple communication task (similar to the one studied by Lazaridou et al., 2016).

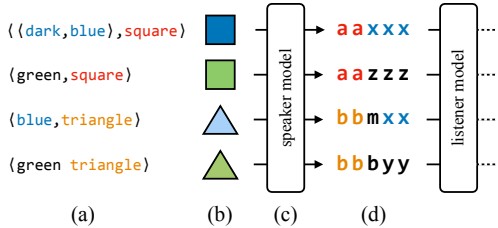

(a)    (b)    (c)    (d)

Figure 1: Representations arising from a communication game. In this game, an observation (b) is presented to a learned speaker model (c), which encodes it as a discrete character sequence (d) to be consumed by a listener model for some downstream task. The space of inputs has known compositional structure (a). We want to measure the extent to which this structure is reflected (perhaps imperfectly) in the structure of the learned codes.

Is this encoding scheme compositional? That is, to what extent can we analyze the agents' messages as being built from smaller pieces (e.g. pieces `xx` meaning *blue* and `bb` meaning *triangle*)?

A large body of work, from early experiments on language evolution to recent deep learning models (Kirby, 1998; Lazaridou et al., 2017), aims to answer questions like this one. But existing solutions rely on manual (and often subjective) analysis of model outputs (Mordatch & Abbeel, 2017), or at best automated procedures tailored to the specifics of individual problem domains (Brighton & Kirby, 2006). They are difficult to compare and difficult to apply systematically.

We are left with a need for a standard, formal, automatable and quantitative technique for evaluating claims about compositional structure in learned representations. The present work aims at first steps toward meeting that need. We focus on an *oracle* setting where the compositional structure of model inputs is known, and where the only question is whether this structure is reflected in model outputs. This oracle evaluation paradigm covers most of the existing representation learning problems in which compositionality has been studied.

The first contribution of this paper is a simple formal framework for measuring how well a collection of representations (discrete- or continuous-valued) reflects an oracle compositional analysis of model inputs. We propose an evaluation metric called TRE, which provides graded judgments of compositionality for a given set of (input, representation) pairs. The core of our proposal is to treat a set of primitive meaning representations as hidden, and optimize over them to find an explicitly compositional model that approximates the true model as well as possible. For example, if the compositional structure that describes an object is a simple conjunction of attributes, we can search for a collection of "attribute vectors" that sum together to produce the observed object representations; if it is a sparse combination of (attribute, value) pairs we can additionally search for "value vectors" and parameters of a binding operation; and so on for more complex compositions.

Having developed a tool for assessing the compositionality of representations, the second contribution of this paper is a survey of applications. We present experiments and analyses aimed at answering four questions about the relationship between compositionality and learning:

- How does compositionality of representations evolve in relation to other measurable model properties over the course of the learning process? (Section 4)
- How well does compositionality of representations track human judgments about the compositionality of model inputs? (Section 5)
- How does compositionality constrain distances between representations, and how does TRE relate to other methods that analyze representations based on similarity? (Section 6)
- Are compositional representations necessary for generalization to out-of-distribution inputs? (Section 7)

We conclude with a discussion of possible applications and generalizations of TRE-based analysis.

## 2 RELATED WORK

Arguments about whether distributed (and other non-symbolic) representations could model compositional phenomena were a staple of 1980s-era connectionist–classicist debates. Smolensky (1991) provides an overview of this discussion and its relation to learnability, as well as a concrete implementation of a compositional encoding scheme with distributed representations. Since then, numerous other approaches for compositional representation learning have been proposed, with (Mitchell & Lapata, 2008; Socher et al., 2012) and without (Dircks & Stoness, 1999; Havrylov & Titov, 2017) the scaffolding of explicit composition operations built into the model.

The main experimental question is thus when and how compositionality arises "from scratch" in the latter class of models. In order to answer this question it is first necessary to determine whether compositional structure is present at all. Most existing proposals come from linguistics and and philosophy, and offer evaluations of compositionality targeted at analysis of formal and natural languages (Carnap, 1937; Lewis, 1976). Techniques from this literature are specialized to the details of linguistic representations—particularly the algebraic structure of grammars (Montague, 1970). It is not straightforward to apply these techniques in more general settings, particularly those featuring non-string-valued representation spaces. We are not aware of existing work that describes a procedure suitable for answering questions about compositionality in the general case.

Machine learning research has responded to this absence in several ways. One class of evaluations (Mordatch & Abbeel, 2017; Choi et al., 2018) derives judgments from ad-hoc manual analyses of representation spaces. These analyses provide insight into the organization of representations but are time-consuming and non-reproducible. Another class of evaluations (Brighton, 2002; Andreas & Klein, 2017; Bogin et al., 2018) exploits task-specific structure (e.g. the ability to elicit pairs of representations known to feature particular relationships) to give evidence of compositionality. Our work aims to provide a standard and scalable alternative to these model- and task-specific evaluations.

Other authors refrain from measuring compositionality directly, and instead base analysis on measurement of related phenomena, for which more standardized evaluations exist. Examples include correlation between representation similarity and similarity of oracle compositional analyses (Brighton & Kirby, 2006) and generalization to structurally novel inputs (Kottur et al., 2017). Our approach makes it possible to examine the circumstances under which these surrogate measures in fact track stricter notions of compositionality; similarity is discussed in Sec. 6 and generalization in Sec. 7.

A long line of work in natural language processing (Coecke et al., 2010; Baroni & Zamparelli, 2010; Clark, 2012; Fyshe et al., 2015) focuses on learning composition functions to produce distributed representations of phrases and sentences—that is, for purposes of modeling rather than evaluation. We use one experiment from this literature to validate our own approach (Section 5). On the whole, we view work on compositional representation learning in NLP as complementary to the framework presented here: our approach is agnostic to the particular choice of composition function, and the aforementioned references provide well-motivated choices suitable for evaluating data from language and other sources. Indeed, one view of the present work is simply as a demonstration that we can take existing NLP techniques for compositional representation learning, fit them to representations produced by other models (even in non-linguistic settings), and view the resulting *training loss* as a measure of the compositionality of the representation system in question.

## 3 EVALUATING COMPOSITIONALITY

Consider again the communication task depicted in Figure 1. Here, a speaker model observes a target object described by a feature vector. The speaker sends a message to a listener model, which uses the message to complete a downstream task—for example, identifying the referent from a collection of distractors based on the content of the message (Enquist & Arak, 1994; Lazaridou et al., 2017). Messages produced by the speaker model serve as *representations* of input objects; we want to know if these representations are compositional. Crucially, we may already know something about the structure of the inputs themselves. In this example, inputs can be identified via composition of categorical shape and color attributes. How might we determine whether this *oracle* analysis of input structure is reflected in the structure of representations? This section proposes an automated procedure for answering the question.

**Representations** A representation learning problem is defined by a dataset $\mathcal{X}$ of *observations* $x$ (Figure 1b); a space $\Theta$ of *representations* $\theta$ (Figure 1d); and a *model* $f : \mathcal{X} \rightarrow \Theta$ (Figure 1c). We assume that the representations produced by $f$ are used in a larger system to accomplish some concrete task, the details of which are not important for our analysis.

**Derivations** The technique we propose additionally assumes we have prior knowledge about the compositional structure of inputs. In particular, we assume that inputs can be labeled with tree-structured *derivations* $d$ (Figure 1a), defined by a finite set $\mathcal{D}_0$ of *primitives* and a binary *bracketing* operation $\langle \cdot, \cdot \rangle$, such that if $d_i$ and $d_j$ are derivations, $\langle d_i, d_j \rangle$ is a derivation. Derivations are produced by a *derivation oracle* $D : \mathcal{X} \rightarrow \mathcal{D}$.

**Compositionality** In intuitive terms, the representations computed by $f$ are compositional if each $f(x)$ is determined by the structure of $D(x)$. Most discussions of compositionality, following Montague (1970), make this precise by defining a *composition* operation $\theta_a * \theta_b \mapsto \theta$ in the space of representations. Then the model $f$ is compositional if it is a homomorphism from inputs to representations: we require that for any $x$ with $D(x) = \langle D(x_a), D(x_b) \rangle$,

$$f(x) = f(x_a) * f(x_b) . \tag{1}$$

In the linguistic contexts for which this definition was originally proposed, it is straightforward to apply. Inputs $x$ are natural language strings. Their associated derivations $D(x)$ are syntax trees, and composition of derivations is syntactic composition. Representations $\theta$ are logical representations of meaning (for an overview see van Benthem & ter Meulen, 1996). To argue that a particular fragment of language is compositional, it is sufficient to exhibit a *lexicon* $\mathcal{D}_0$ mapping words to their associated meaning representations, and a *grammar* for composing meanings where licensed by derivations. Algorithms for learning grammars and lexicons from data are a mainstay of semantic

parsing approaches to language understanding problems like question answering and instruction following (Zettlemoyer & Collins, 2005; Chen, 2012; Artzi et al., 2014).

But for questions of compositionality involving more general representation spaces and more general analyses, the above definition presents two difficulties: (1) In the absence of a clearly-defined syntax of the kind available in natural language, how do we identify lexicon entries: the primitive parts from which representations are constructed? (2) What do we do with languages like the one in Figure 1d, which seem to exhibit *some* kind of regular structure, but for which the homomorphism condition given in Equation 1 cannot be made to hold exactly?

Consider again the example in Figure 1. The oracle derivations tell us to identify primitive representations for *dark*, *blue*, *green*, *square*, and *triangle*. The derivations then suggest a process for composing these primitives (e.g. via string concatenation) to produce full representations. The speaker model is compositional (in the sense of Equation 1) as long as there is *some* assignment of representations to primitives such that for each model input, composing primitive representations according to the oracle derivation reproduces the speaker's prediction.

In Figure 1 there is no assignment of strings to primitives that reproduces model predictions exactly. But predictions can be reproduced approximately—by taking xx to mean *blue*, aa to mean *square*, etc. The quality of the approximation serves as a measure of the compositionality of the true predictor: predictors that are mostly compositional but for a few exceptions, or compositional but for the addition of some noise, will be well-approximated on average, while arbitrary mappings from inputs to representations will not. This suggests that we should measure compositionality by searching for representations that allow an explicitly compositional model to approximate the true $f$ as closely as possible. We define our evaluation procedure as follows:

---

**Tree Reconstruction Error (TRE)**

First choose :

- a distance function $\delta : \Theta \times \Theta \to [0, \infty)$ satisfying $\delta(\theta, \theta') = 0 \Leftrightarrow \theta = \theta'$
- a composition function $* : \Theta \times \Theta \to \Theta$

Define $\hat{f}_\eta(d)$, a *compositional approximation to $f$* with parameters $\eta$, as:

$$\hat{f}_\eta(d_i) = \eta_i \qquad \text{for } d_i \in \mathcal{D}_0$$
$$\hat{f}_\eta(\langle d, d' \rangle) = \hat{f}_\eta(d) * \hat{f}_\eta(d') \qquad \text{for all other } d$$

$\hat{f}_\eta$ has one parameter vector $\eta_i$ for every $d_i$ in $\mathcal{D}_0$; these vectors are members of the representation space $\Theta$.

Given a dataset $\mathcal{X}$ of inputs $x_i$ with derivations $d_i = D(x_i)$, compute:

$$\eta^* = \arg\min_\eta \sum_i \delta\big(f(x_i), \hat{f}_\eta(d_i)\big) \qquad (2)$$

Then we can define datum- and dataset-level evaluation metrics:

$$\text{TRE}(x) = \delta\big(f(x), \hat{f}_{\eta^*}(d)\big) \qquad (3)$$
$$\text{TRE}(\mathcal{X}) = \frac{1}{n} \sum_i \text{TRE}(x_i) \qquad (4)$$

---

**TRE and compositionality** How well does the evaluation metric $\text{TRE}(\mathcal{X})$ capture the intuition behind Equation 1? The definition above uses parameters $\eta_i$ to witness the constructability of representations from parts, in this case by explicitly optimizing over those parts rather than taking them to be given by $f$. Each term in Equation 2 is analogous to an instance of Equation 1, measuring how well $\hat{f}_{\eta^*}(x_i)$, the best compositional prediction, matches the true model prediction $f(x_i)$. In the case of models that are homomorphisms in the sense of Equation 1, TRE reduces to the familiar case:

**Remark 1.** $\mathrm{TRE}(x) = 0$ *for all $x$ if and only if Equation 1 holds exactly (that is, $f(x) = f(x_a) * f(x_b)$ for any $x, x_a, x_b$ with $D(x) = \langle D(x_a), D(x_b) \rangle$).*

*Proof.* One direction follows immediately from defining $\hat{f}_{\eta^*}(x) = f(x)$. For the other, $f(x) = \hat{f}(D(x)) = \hat{f}(\langle D(x_a), D(x_b) \rangle) = \hat{f}(D(x_a)) * \hat{f}(D(x_b)) = f(x_a) * f(x_b)$. $\qquad\square$

**Learnable composition operators**   The definition of TRE leaves the choice of $\delta$ and $*$ up to the evaluator. Indeed, if the exact form of the composition function is not known *a priori*, it is natural to define $*$ with free parameters (as in e.g. Baroni & Zamparelli, 2010), treat these as another learned part of $\hat{f}$, and optimize them jointly with the $\eta_i$. However, some care must be taken when choosing $*$ (especially when learning it) to avoid trivial solutions:

**Remark 2.** *Suppose $D$ is injective; that is, every $x \in \mathcal{X}$ is assigned a unique derivation. Then there is always some $*$ that achieves $\mathrm{TRE}(\mathcal{X}) = 0$: simply* define $f(x_a) * f(x_b) = f(x)$ *for any $x, x_a, x_b$ as in the preceding definition, and set $\hat{f} = f$.*

In other words, some pre-commitment to a restricted composition function is essentially inevitable: if we allow the evaluation procedure to select an arbitrary composition function, the result will be trivial. This paper features experiments with $*$ in both a fixed functional form and a learned parametric one.

**Implementation details**   For models with continuous $\Theta$ and differentiable $\delta$ and $*$, $\mathrm{TRE}(\mathcal{X})$ is also differentiable. Equation 2 can be solved using gradient descent. We use this strategy in Sections 4 and 5. For discrete $\Theta$, it may be possible to find a continuous *relaxation* with respect to which $\delta(\theta, \cdot)$ and $*$ are differentiable, and gradient descent again employed. We use this strategy in Section 7 (discussed further there). An implementation of an SGD-based TRE solver is provided in the accompanying software release. For other problems, task-specific optimizers (e.g. machine translation alignment models; Bogin et al., 2018) or general-purpose discrete optimization toolkits can be applied to Equation 2.

The remainder of the paper highlights ways of using TRE to answer questions about compositionality that arise in machine learning problems of various kinds.

## 4   COMPOSITIONALITY AND LEARNING DYNAMICS

We begin by studying the relationship between compositionality and learning dynamics, focusing on the information bottleneck theory of representation learning proposed by Tishby & Zaslavsky (2015). This framework proposes that learning in deep models consists of an error minimization phase followed by a compression phase, and that compression is characterized by a decrease in the mutual information between inputs and their computed representations. We investigate the hypothesis that the compression phase finds a *compositional* representation of the input distribution, isolating decision-relevant attributes and discarding irrelevant information.

Data comes from a few-shot classification task. Because our analysis focuses on compositional hypothesis classes, we use visual concepts from the Color MNIST dataset of Seo et al. (2017) (Figure 2). We predict classifiers in a meta-learning framework (Schmidhuber, 1987; Santoro et al., 2016): for each sub-task, the learner is presented with two images corresponding to some compositional visual concept (e.g. "digit 8 on a black background" or "green with heavy stroke") and must determine whether a held-out image is an example of the same visual concept.

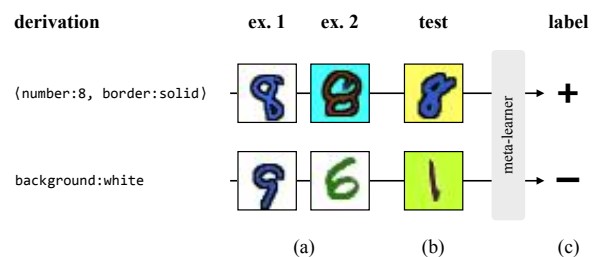

Figure 2: Meta-learning task: learners are presented with two example images depicting a visual concept (a), and must determine whether a third image (b) is an example of the same concept (c).

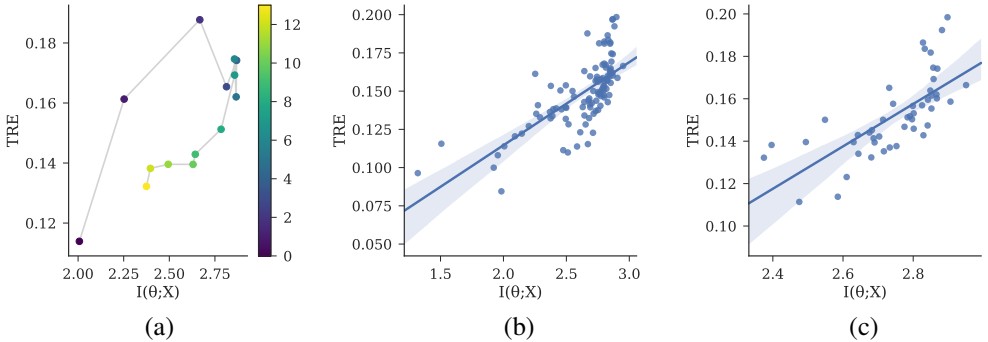

(a)  (b)  (c)

Figure 3: Relationship between reconstruction error TRE and mutual information $I(\theta; X)$ between inputs and representations. (a) Evolution of the two quantities over the course of a single run. Both initially increase, then decrease. The color bar shows the training epoch. (b) Values from ten training runs. (c) Values from the second half of each training run, taken to begin when $I(\theta; X)$ reaches a maximum. In (b) and (c), the observed correlation is significant: respectively ($r = 0.70$, $p < 1e{-}10$) and ($r = 0.71$, $p < 1e{-}8$).

Given example images $x_1$ and $x_2$, a test image $x^*$, and label $y^*$, the model computes:

$$z_i = \texttt{CNN}(x_i) \text{ for } i \in \{1, 2, *\}$$
$$\theta = \tanh(W(z_1 + z_2))$$
$$\hat{y} = \theta^\top z_t$$

We use $\theta$ as the representation of a classifier for analysis. The model is trained to minimize the logistic loss between logits $\hat{y}$ and ground-truth labels $y^*$. More details are given in Appendix A.

**Compositional structure**   Visual concepts used in this task are all single attributes or conjunctions of attributes; i.e. their associated derivations are of the form $\texttt{attr}$ or $\langle \texttt{attr}_1, \texttt{attr}_2 \rangle$. Attributes include background color, digit color, digit identity and stroke type. The composition function $*$ is addition and the distance $\delta(\theta, \theta')$ is cosine similarity $1 - \theta^\top \theta' / (\|\theta\| \|\theta'\|)$.

**Evaluation**   The training dataset consists of 9000 image triplets, evenly balanced between positive and negative classes, with a validation set of 500 examples. At convergence, the model achieves validation accuracy of 75.2% on average over ten training runs. (Perfect accuracy is not possible because the true classifier is not fully determined by two training examples). We explore the relationship between the information bottleneck and compositionality by comparing $\text{TRE}(\mathcal{X})$ to the mutual information $I(\theta; x)$ between representations and inputs over the course of training. Both quantities are computed on the validation set, calculating $\text{TRE}(\mathcal{X})$ as described in Section 3 and $I(\theta; X)$ as described in Shwartz-Ziv & Tishby (2017). (For discussion of limitations of this approach to computing mutual information between inputs and representations, see Saxe et al., 2018.)

Figure 3 shows the relationship between $\text{TRE}(\mathcal{X})$ and $I(\theta; X)$. Recall that small TRE is indicative of a high degree of compositionality. It can be seen that both mutual information and reconstruction error are initially low (because representations initially encode little about distinctions between inputs). Both increase over the course of training, and decrease together after mutual information reaches a maximum (Figure 3a). This pattern holds if we plot values from multiple training runs at the same time (Figure 3b), or if we consider only the postulated compression phase (Figure 3c). These results are consistent with the hypothesis that compression in the information bottleneck framework is associated with the discovery of compositional representations.

## 5   COMPOSITIONALITY AND HUMAN JUDGMENTS

Next we investigate a more conventional representation learning task. High-dimensional embeddings of words and phrases are useful for many natural language processing applications (Turian et al., 2010), and many techniques exist to learn them from unlabeled text (Deerwester et al., 1990; Mikolov et al., 2013). The question we wish to explore is not whether phrase vectors are compositional in

aggregate, but rather how compositional individual phrase representations are. Our hypothesis is that bigrams whose representations have low TRE are those whose *meaning* is essentially compositional, and well-explained by the constituent words, while bigrams with large reconstruction error will correspond to non-compositional multi-word expressions (Nattinger & DeCarrico, 1992).

This task is already well-studied in the natural language processing literature (Salehi et al., 2015), and the analysis we present differs only in the use of TRE to search for atomic representations rather than taking them to be given by pre-trained word representations. Our goal is to validate our approach in a language processing context, and show how existing work on compositionality (and representations of natural language in particular) fit into the more general framework proposed in the current paper.

We train embeddings for words and bigrams using the CBOW objective of Mikolov et al. (2013) using the implementation provided in FastText (Bojanowski et al., 2017) with 100-dimensional vectors and a context size of 5. Vectors are estimated from a 250M-word subset of the Gigaword dataset (Parker et al., 2011). More details are provided in Appendix A.

**Compositional structure**   We want to know how close phrase embeddings are to the composition of their constituent word embeddings. We define derivations for words and phrases in the natural way: single words $w$ have primitive derivations $d = w$; bigrams $w_1 w_2$ have derivations of the form $\langle w_1, w_2 \rangle$. The composition function is again vector addition and distance is cosine distance. (Future work might explore learned composition functions as in e.g. Grefenstette et al., 2013, for future work.) We compare bigram-level judgments of compositionality computed by TRE with a dataset of human judgments about noun–noun compounds (Reddy et al., 2011). In this dataset, humans rate bigrams as compositional on a scale from 0 to 5, with highly conventionalized phrases like *gravy train* assigned low scores and *graduate student* assigned high ones.

**Results**   We reproduce the results of Salehi et al. (2015) within the tree reconstruction error framework: for a given $x$, TRE($x$) is anticorrelated with human judgments of compositionality ($\rho = -0.34$, $p < 0.01$). Collocations rated "most compositional" by our approach (i.e. with lowest TRE) are: *application form*, *polo shirt*, *research project*; words rated "least compositional" are *fine line*, *lip service*, and *nest egg*.

## 6   COMPOSITIONALITY AND SIMILARITY

The next section aims at providing a formal, rather than experimental, characterization of the relationship between TRE and another perspective on the analysis of representations with help from oracle derivations. Brighton & Kirby (2006) introduce a notion of *topographic similarity*, arguing that a learned representation captures relevant domain structure if distances between learned representations are correlated with distances between their associated derivations. This can be viewed as providing a weak form of evidence for compositionality—if the distance function rewards pairs of representations that share overlapping substructure (as might be the case with e.g. string edit distance), edit distance will be expected to correlate with some notion of derivational similarity (Lazaridou et al., 2018).

In this section we aim to clarify the relationship between the two evaluations. To do this we first need to equip the space of derivations described in Section 3 with a distance function. As the derivations considered in this paper are all tree-structured, it is natural to use a simple *tree edit distance* (Bille, 2005) for this purpose. We claim the following:

**Proposition 1.** *Let $\hat{f} = \hat{f}_{\eta^*}$ be an approximation to $f$ estimated as in Equation 2, with all TRE($x$) $\leq \epsilon$ for some $\epsilon$. Let $\Delta$ be the tree edit distance (defined formally in Appendix B, Definition 2), and let $\delta$ be any distance on $\Theta$ satisfying the following properties:*

1. *$\delta(\hat{f}(d_i), \hat{f}(d_j)) \leq 1$ for $d_i, d_j \in \mathcal{D}_0$*

2. *$\delta(\hat{f}(d), 0) \leq 1$ for $d \in \mathcal{D}_0$, where $0$ is the identity element for $*$.*

3. *$\delta(\theta_i * \theta_j, \theta_k * \theta_\ell) \leq \delta(\theta_i, \theta_k) + \delta(\theta_j, \theta_\ell)$.*
   *(This condition is satisfied by any translation-invariant metric.)*

*Then $\Delta$ is an approximate upper bound on $\delta$: for any $x$, $x'$ with $d = D(x)$, $d' = D(x')$,*

$$\delta(f(x), f(x')) \leq \Delta(d, d') + 2\epsilon . \tag{5}$$

In other words, representations cannot be much farther apart than the derivations that produce them. Proof is provided in Appendix B.

We emphasize that small TRE is not a sufficient condition for topographic similarity as defined by Brighton & Kirby (2006): very different derivations might be associated with the same representation (e.g. when representing arithmetic expressions by their results). But this result does demonstrate that compositionality imposes some constraints on the inferences that can be drawn from similarity judgments between representations.

## 7 COMPOSITIONALITY AND GENERALIZATION

In our final set of experiments, we investigate the relationship between compositionality and generalization. Here we focus on communication games like the one depicted in Figure 1 and in more detail in Figure 4. As in the previous section, existing work argues for a relationship between compositionality and generalization, claiming that agents need compositional communication protocols to generalize to unseen referents (Kottur et al., 2017; Choi et al., 2018). Here we are able to evaluate this claim empirically by training a large number of agents from random initial conditions, measuring the compositional structure of the language that emerges, and seeing how this relates to their performance on both familiar and novel objects.

Our experiment focuses on a *reference game* (Gatt et al., 2007). Two policies are trained: a speaker and a listener. The speaker observes pair of *target* objects represented with a feature vector. The speaker then sends a message (coded as a discrete character sequence) to the listener model. The listener observes this message and attempts to reconstruct the target objects by predicting a sequence of attribute sets. If all objects are predicted correctly, both the speaker and the listener receive a reward of 1 (partial credit is awarded for partly-correct objects; Figure 4).

Because the communication protocol is discrete, policies are jointly trained using a policy gradient objective (Williams, 1992). The speaker and listener are implemented with RNNs; details are provided in Appendix A.

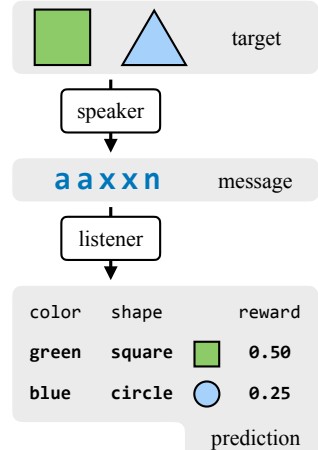

Figure 4: The communication task: A *speaker* model observes a pair of target objects, and sends a description of the objects (as a discrete code) to a *listener* model. The listener attempts to reconstruct the targets, receiving fractional reward for partially-correct predictions.

**Compositional structure** Every target referent consists of two objects; each object has two attributes. The derivation associated with each communicative task thus has the tree structure $\langle\langle \text{attr}_{1a}, \text{attr}_{1b}\rangle, \langle \text{attr}_{2a}, \text{attr}_{2b}\rangle\rangle$. We *hold out* a subset of these object pairs at training time to evaluate generalization: in each training run, 1/3 of possible reference candidates are never presented to the agent at training time.

Where the previous examples involved a representation space of real embeddings, here representations are fixed-length discrete codes. Moreover, the derivations themselves have a more complicated semantics than in Sections 4 and 5: order matters, and a commutative operation like addition cannot capture the distinction between $\langle\langle \text{green}, \text{square}\rangle, \langle \text{blue}, \text{triangle}\rangle\rangle$ and $\langle\langle \text{green}, \text{triangle}\rangle, \langle \text{blue}, \text{square}\rangle\rangle$. We thus need a different class of composition and distance operations. We represent each agent message as a sequence of one-hot vectors, and take the error function $\delta$ to be the $\ell_1$ distance between vectors. The composition function has the form:

$$\theta * \theta' = A\theta + B\theta' \tag{6}$$

with free composition parameters $\eta_* = \{A, B\}$ in Equation 2. These matrices can redistribute the tokens in $\theta$ and $\theta'$ across different positions of the input string, but cannot affect the choice of the tokens themselves; this makes it possible to model non-commutative aspects of string production. To

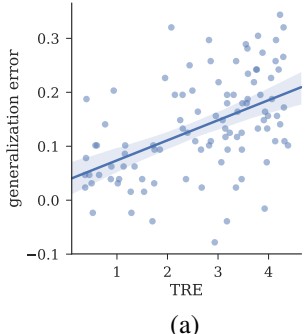
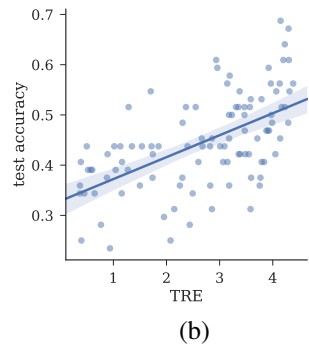

(a)                                    (b)

Figure 5:    Relationship between TRE and reward. (a) Compositional languages exhibit lower generalization error, measured as the difference between train and test reward ($r = 0.50$, $p < 1e-6$). (b) However, compositional languages also exhibit lower absolute performance ($r = 0.57$, $p < 1e-9$). Both facts remain true even if we restrict analysis to "successful" training runs in which agents achieve a reward $> 0.5$ on held-out referents ($r = 0.6$, $p < 1e-3$ and $r = 0.38$, $p < 0.05$ respectively).

|  | Language A | Language B |
|---|---|---|
| ((red circle) (blue triangle)) | jjjj | jeoo |
| ((red circle) (blue star)) | oppp | jjjj |
| ((red circle) (blue circle)) | oopp | jjjj |
| ((red circle) (blue square)) | oopp | jjjb |
| ((red square) (blue triangle)) | jjjj | jbjj |
| ((red square) (blue star)) | oooo | jbjj |
| ((red square) (blue circle)) | oooo | jbbb |
| ((red square) (blue square)) | oooo | jbbb |
| TRE | 4.30 | 2.96 |
| Train reward | 0.78 | 0.75 |
| Test reward | 0.61 | 0.59 |

Figure 6:  Fragment of languages resulting from two multiagent training runs. In the first section, the left column shows the target referent, while the remaining columns show the message generated by speaker in the given training run after observing the referent. The two languages have substantially different TRE, but induce similar listener performance (*Train* and *Test reward*).

compute TRE via gradient descent, we allow the elements of $\mathcal{D}_0$ to be arbitrary vectors (intuitively assigning fractional token counts to string indices) rather than restricting them to one-hot indicators. With this change, both $\delta$ and $*$ have subgradients and can be optimized using the same procedure as in preceding sections.

**Results**    We train 100 speaker–listener pairs with random initial parameters and measure their performance on both training and test sets. Our results suggest a more nuanced view of the relationship between compositionality and generalization than has been argued in the existing literature. TRE is significantly correlated with generalization error (measured as the difference between training accuracies, Figure 5a). However, TRE is also significantly correlated with absolute model reward (Figure 5b)—"compositional" languages more often result from poor communication strategies than successful ones. This is largely a consequence of the fact that many languages with low TRE correspond to trivial strategies (for example, one in which the speaker sends the same message regardless of its observation) that result in poor overall performance.

Moreover, despite the correlation between TRE and generalization error, low TRE is by no means a necessary condition for good generalization. We can use our technique to automatically mine a collection of training runs for languages that achieve good generalization performance at both low and high levels of compositionality. Examples of such languages are shown in Figure 6.

## 8    CONCLUSIONS

We have introduced a new evaluation method called TRE for generating graded judgments about compositional structure in representation learning problems where the structure of the observations is understood. TRE infers a set of primitive meaning representations that, when composed, approximate the observed representations, then measures the quality of this approximation. We have applied TRE-based analysis to four different problems in representation learning, relating compositionality to learning dynamics, linguistic compositionality, similarity and generalization.

Many interesting questions regarding compositionality and representation learning remain open. The most immediate is how to generalize TRE to the setting where oracle derivations are not available; in this case Equation 2 must be solved jointly with an unsupervised grammar induction problem (Klein & Manning, 2004). Beyond this, it is our hope that this line of research opens up two different kinds of new work: better understanding of existing machine learning *models*, by providing a new set of tools for understanding their representational capacity; and better understanding of *problems*, by better understanding the kinds of data distributions and loss functions that give rise to compositional- or non-compositional representations of observations.

REPRODUCIBILITY

Code and data for all experiments in this paper are provided at
`https://github.com/jacobandreas/tre`.

ACKNOWLEDGMENTS

Thanks to Daniel Fried and David Gaddy for feedback on an early draft of this paper. The author was supported by a Facebook Graduate Fellowship at the time of writing.

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

## A    MODELING DETAILS

**Few-shot classification**    The CNN has the following form:

```
Conv(out=6, kernel=5)
ReLU
MaxPool(kernel=2)
Conv(out=16, kernel=5)
ReLU
MaxPool(kernel=2)
Linear(out=128)
ReLU
Linear(out=64)
ReLU
```

The model is trained using ADAM (Kingma & Ba, 2014) with a learning rate of .001 and a batch size of 128. Training is ended when the model stops improving on a held-out set.

**Word embeddings**    We train FastText (Bojanowski et al., 2017) on the first 250 million words of the NYT section of Gigaword (Parker et al., 2011). To acquire bigram representations, we pre-process this dataset so that each occurrence of a bigram from the Reddy et al. (2011) dataset is treated as a single word for purposes of estimating word vectors.

**Communication**    The encoder and decoder RNNs both use gated recurrent units (Cho et al., 2014) with embeddings and hidden states of size 256. The size of the discrete vocabulary is set to 16 and the maximum message length to 4. Training uses a policy gradient objective with a scalar baseline set to the running average reward; this is optimized using ADAM (Kingma & Ba, 2014) with a learning rate of .001 and a batch size of 256. Each model is trained for 500 steps. Models are trained by sampling from the decoder's output distribution, but greedy decoding is used to evaluate performance and produce Figure 6.

## B    PROPOSITION 1

First, some definitions:

**Definition 1.** *The **size** of a derivation is given by:*

$$|d| = 1 \qquad \text{if } d \in \mathcal{D}_0$$
$$|\langle d_a, d_b \rangle| = |d_a| + |d_b| \qquad \text{otherwise} \tag{7}$$

**Definition 2.** *The **tree edit distance** between derivations is defined by:*

$$\Delta(d_i, d_j) = \mathbb{I}[i = j] \quad \text{if } d_i \in \mathcal{D}_0 \text{ and } d_j \in \mathcal{D}_0$$

$$\Delta(d_i, \langle d_j, d_k \rangle) = \min \left\{ \begin{array}{l} \Delta(d_i, d_j) + |d_k| \\ \Delta(d_i, d_k) + |d_j| \end{array} \right\} \quad \text{if } d_i \in \mathcal{D}_0$$

$$\Delta(\langle d_i, d_j \rangle, \langle d_k, d_\ell \rangle) = \min \left\{ \begin{array}{ll} \Delta(d_i, d_k) + \Delta(d_j, d_\ell) \\ \Delta(\langle d_i, d_j \rangle, d_k) + |d_\ell| & \Delta(\langle d_i, d_j \rangle, d_\ell) + |d_k| \\ \Delta(\langle d_k, d_\ell \rangle, d_i) + |d_j| & \Delta(\langle d_k, d_\ell \rangle, d_j) + |d_i| \end{array} \right\} \tag{8}$$

Now, suppose we have $x$ and $x'$ with derivations $d = D(x)$, $d' = D(x')$ and representations $\theta = f(x)$, $\theta' = f(x')$. Proposition 1 claims that $\delta(\theta, \theta') \leq \Delta(d, d') + 2\epsilon$.

**Lemma 1.** $\delta(\hat{f}(d), 0) \leq |d|$.

*Proof.* For $d \in \mathcal{D}_0$ this follows immediately from Condition 2 in the proposition. For composed derivations it follows from Condition 3 taking $\theta_k = \theta_\ell = 0$ and induction on $|d|$. □

**Lemma 2.** $\delta(\hat{f}(d), \hat{f}(d')) \leq \Delta(d, d')$

*Proof.* By induction on the structure of $d$ and $d'$:

**Base case** Both $d, d' \in \mathcal{D}_0$.

If $d = d'$, $\delta(\hat{f}(d), \hat{f}(d')) = \delta(\hat{f}(d), \hat{f}(d)) = 0 = \Delta(d, d')$.

If $d \neq d'$, $\delta(\hat{f}(d), \hat{f}(d')) \leq 1 = \Delta(d, d')$ from Condition 1.

**Inductive case** Consider the arrangement of derivations that minimizes Equation 8 for derivation $d$ and $d'$. There are two possibilities:

*Case 1:* $\Delta(d, d')$ has the form $\Delta(d_i, d_k) + \Delta(d_j, d_\ell)$ for some $d_{i,j,k,\ell}$. W.l.o.g. let $d = \langle d_i, d_j \rangle$ and $d' = \langle d_k, d_\ell \rangle$. Then,

$$
\begin{aligned}
\delta(\hat{f}(d), \hat{f}(d')) &= \delta(\hat{f}(d_i) * \hat{f}(d_j), \hat{f}(d_k) * \hat{f}d_\ell) \\
&\leq \delta(\hat{f}(d_i), \hat{f}(d_k)) + \delta(\hat{f}(d_j), \hat{f}d_\ell) \\
&\leq \Delta(d_i, d_k) + \Delta(d_j, d_\ell) \\
&= \Delta(d, d')
\end{aligned}
$$

*Case 2:* $\Delta(d, d')$ has the form $\Delta(d_i, d_k) + |d_j|$ for some $d_{i,j,k}$. W.l.o.g. let $d = \langle d_i, d_j \rangle$ and $d' = d_k$. Abusing notation slightly, let us define $\Delta(d, 0) = |d|$. If we let $d_\ell = 0$ this case reduces to the previous one. $\qquad\square$

Finally,

*Proof of Proposition 1.*

$$
\begin{aligned}
\delta(\theta, \theta') &\leq \delta(\hat{f}(d), \hat{f}(d')) + 2\epsilon \\
&\leq \Delta(d, d') + 2\epsilon
\end{aligned}
$$

$\qquad\square$

