# OpenReview forum: "Measuring Compositionality in Representation Learning"
_ICLR.cc/2019/Conference_

### Official Review · AnonReviewer3 · 2018-11-01
**Refreshingly pedagogical paper; limited experiments despite asking some really interesting questions**

**Rating:** 7
**Confidence:** 4

**Review:**

The paper tackles a very interesting problem about representations, especially of the connectionist kind -- how do we know if the learned representations capture the compositional structure present in the inputs, and tries to come up with a systematic framework to answer that question. The framework assumes the presence of an oracle that can give us the true compositional structure. Then the author try to answer some refreshing questions about the dynamics of learning and compositionality while citing some interesting background reading.

However, I’m a bit torn about the experiments. On the one hand, I like the pedagogical nature of the experiments. They are small and should be easy to reproduce. On the other hand, all of them seem to be fairly similar kinds of composition with very few attributes (mostly bigrams). So whether the intuitions hold for more complex compositional structures is hard to say.

Nevertheless, it’s a well written paper and is a helpful first step towards studying the problem of compositionality in vector representations.


Minor points
Pg 3 “_grammar_ for composing meanings *where* licensed by derivations” seems incorrect.
Figure 5: seems quite noisy to make the linear relationship claim

EDIT: I still think the compositions under consideration are the simpler ones. Still with the new experiments the coverage seems nicer. Given the authors plan to release their source code, I expect there will be an opportunity for the rest of the community to build on these, to test TRE's efficacy on more complex compositions. I updated my scores to reflect the change.

---

> ### Author Response · Authors · 2018-11-18
> **Response**
>
> Thanks for your review---we hope the new experiments address your concerns about the generalization to new kinds of composition functions and deeper trees. Regarding Fig. 5---the relationship is indeed noisy, but as discussed in the paper the correlation is measurable and statistically significant to a high degree.

---

### Official Review · AnonReviewer2 · 2018-11-03

**Rating:** 6
**Confidence:** 4

**Review:**

Edit and a further question: Reading again Section 7, I'm wondering whether the  the high generalization is possible due to the fact that at test time only one of the two candidates is unseen, and the other is seen. Having *both* candidates to be unseen makes the problem significantly harder since the only way for the listener to get it right is to associate the message with the right candidate, rather than relying in some other strategy like whether the message is novel (thus it's the seen candidate) or new (thus it's the unseen candidate). As such, I don't think I can fully trust your conclusions due to this potential confounder.
--------------------------------------------------------------

The authors propose a measure of compositionality in representations. Given instances of data x annotated with semantic primitives, the authors learn a vector for each of the primitive such that the addition of the vectors of the primitives is very close (in terms of cosine) to the latent representation  z of the input x. The authors find that this measure correlates with the mutual information between the input x and z, approximates the human judges of compositionality on a language dataset and finally presents a study on the relation between the proposed measure and generalizalization performance, concluding that their measure correlates with generalization error as well as absolute test accuracy.

This in an interesting study and attacks a very fundamental question; tracking compositionality in representations could pave the way towards representations that facilitate transfer learning and better generalization. While the paper is very clear with respects to results, I found the presentation of the proposed measure overly confusing (and somewhat more exaggerated that what is really going on).

The authors start with a very clean example, that can potentially facilitate clarifying in a visual way the process of obtaining the measure. However, I feel that clarity is being traded-off for formality. It needs several reads to really distill the idea that essentially the authors are simply learning vectors of primitives that when added should resemble the representation of the input. Moreover, the name of the measure is a bit misleading and not justified by the experiments and the data. The authors do not deal with trees in any of the examples, but rather with a set of primitives (apparent in the use of addition as a composition function which being commutative does not allow for word-order and the like deep syntactic properties).

Now, onto the measure. I like the idea of learning basis vectors from the representations and constraining to follow the primitive semantics. Of course, this constraints quite a bit the form of compositionality that the authors are searching for.
The idea of additive semantics has been explored in NLP, however it's mostly applicable for primitives with intersective semantics (e.g., a white towel is something that is both white and a towel). Do the authors think that this restricts their experiments (especially the natural languages ones)? What about other composition techniques found in the literature of compositional semantics (e.g., by Baroni and Zamparelli, 2010).
This is good to be clarified.  Moreover, given the simplicity of the datasets in the current study, wouldn't a reasonable baseline be to obtain the basis vector of blue by averaging all the latent representations of blue?  Similarly, how sensitive are conclusions with respect to different composition functions?

Section 4 is potentially very interesting, but I don't seem to understand why it's good news that TRE correlates with I(x;\theta). Low TRE indicates high-degree of compositionality. I suspect that low MI means that input and latent representation are somewhat independent but I don't see the connection to compositional components. Can the authors clarify?

Section 5 is a nice addition. The authors mention that they learn word and phrase representations. Where are the word representations used? My understanding is that you derive basis word representations by using SGD and the phrase vectors and compute TRE with these. If this is the case, an interesting experiment would be to report how similar the induced basis vectors are (either some first-order or second-order similarity) to the pre-trained ones.

Section 8 presents results on discrete representations. Since this is the experiment most similar to the recent work that uses topographic similarity (and since the authors already prime the reader at section 7 about relation between the 2 measures), it would be interesting to see the empirical relation between TRE and topographic and its relation to generalization and absolute performance.

Baroni and Zamparelli (2010) Nouns are vectors, adjectives are matrices: Representing adjective-noun constructions in semantic space

---

> ### Author Response · Authors · 2018-11-18
> **Response**
>
> - Thank you for the citation suggestion---as mentioned in the response to R1, we've expanded the related work section to talk about this general line of NLP work, and mentioned in a couple of other places where learned "NLP"-style composition functions could be used. One could implement the specific model from B&Z in our framework (modulo some rank constraints) by taking the composition function to be bilinear with a particular choice of tensor.
>
> - Re section 4: the intuition is that there's lots of information in the input images that is not part of the compositional analysis and not relevant for the eventual classification (e.g. the saturation of the pixel in the top-left corner of the image). A good representation will abstract out this irrelevant information, thus reducing MI between the representation and the input image. The experiments in this section suggest that this process of abstraction also results in more compositional representations.
>
> - Re section 5: the single-word representations are also included in the representation dataset X---that is, the model needs to find primitives that are both close to single-word representations and compose properly.
>
> - If time permits, we will try to add extra topographic similarity experiments to S.7.

---

### Official Review · AnonReviewer1 · 2018-11-03
**Solid paper on an interesting topic - some questions as to generalization in other settings**

**Rating:** 6
**Confidence:** 4

**Review:**

This paper describes a framework - Tree Reconstruction Error (TRE) - for assessing compositionality of representations by comparing the learned outputs against those of the closest compositional approximation. The paper demonstrates the use of this framework to assess the role of compositionality in a hypothetical compression phase of representation learning, compares the correspondence of TRE with human judgments of compositionality of bigrams, provides an explanation of the relationship of the metric to topographic similarity, and uses the framework to draw conclusions about the role of compositionality in model generalization.

Overall I think this is a solid paper, with an interesting and reasonable approach to quantifying compositionality, and a fairly compelling set of results. The reported experiments cover reasonable ground in terms of questions relevant to compositionality (relationship to representation compression, generalization), and I appreciate the comparison to human judgments, which lends credibility to applicability of the framework. The results are generally intuitive and reasonable enough to be credible as indicators of how compositionality relates to aspects of learning, while providing some potential insight. The paper is clearly written, and to my knowledge the approach is novel.

I would say the main limitation to the conclusions that can be drawn from these experiments lies in the necessity of committing to a particular composition operator, of which the authors have selected very simple ones without comparing to others. There is nothing obviously unreasonable about the choices of composition operator, but it seems that the conclusions drawn cannot be construed to apply to compositionality as a general concept, but rather to compositionality when defined by these particular operators. Similar limitations apply to the fact that the tests have been run on very specific tasks - it is not clear how these conclusions would generalize to other tasks.

Despite this limitation, I'm inclined to say that the introduction of the framework is a solid contribution, and the results presented are interesting. I think this is a reasonable paper to accept for publication.

Minor comment:
p8 typo: "training and accuracies"

------

Reviewer 2 makes a good point that the presentation of the framework could be much clearer, currently obscuring the central role of learning the primitive representations. This is something that would benefit from revision. Reviewer 2's comments also remind me that, from a perspective of learning composition-ready primitives, Fyshe et al. (2015) is a relevant reference here, as it similarly learns primitive (word) representations to be compatible with a chosen composition function.

Beyond issues of presentation, it seems that we are all in agreement that the paper's takeaways would also benefit from an increase in the scope of the experiments. I'm happy to adjust my score to reflect this.

Reference:
Fyshe et al. (2015) A compositional and interpretable semantic space.

---

> ### Author Response · Authors · 2018-11-18
> **Response**
>
> Thanks for the suggested Fyshe cite---we've realized that the related work section should really spend more time on the (large collection) of NLP papers about learning composition functions to predict phrase representations. We've updated the paper accordingly.

---

### Author Response · Authors · 2018-11-18
**General response**

Dear reviewers,

Thank you all for your detailed feedback! We are glad that you found our submission to be an "interesting" and "pedagogical" study of a "fundamental question". All of the reviews touched on a similar set of points, so we're addressing most of them in this top-level comment and will reply to individual reviews about more specific questions.

First off: we've re-worked the communication experiments in section 7 in response to reviewer feedback. A new paper draft containing these changes has been uploaded. Briefly, the experiment now features:

- a more complicated set of referents (with real tree structures of the form <<obj1:attr1, obj1:attr2>, <obj2:attr1, obj2:attr2>>)
- a more interesting composition function (which is learned, non-commutative and sensitive to string indices)
- as suggested by R2, a more challenging task for the listener (which is now required to generate referents rather than simply recognize them)

The high-level experimental conclusions have remained essentially the same---the only real difference is that some of the correlations are stronger than observed in the initial experiments.

We really appreciate the suggestions that led to these changes, and believe the new experiments better exercise all the pieces of the TRE framework. We hope they also address the main points raised in all the reviews, namely:

- Choice of composition and distance functions: as mentioned, the new version of Sec. 7 has a new (learned, non-commutative) composition function, and continues to use l1 distance for similarity. This means the four example applications in the paper now feature 3 kinds of composition function (addition, the learned linear operation of S.7, and the general class considered in S.6), and 3 kinds of distance function (cosine similarity, l1 distance, and the general class in S.6). Between this and the fact that experiments cover examples from computer vision, natural language processing, and multiagent reinforcement learning, we believe we have provided fairly comprehensive evidence for the generality of TRE.

- Necessity of pre-selecting a specific composition function: First, we emphasize that this choice is also a feature of all previous work that has attempted to analyze compositionality---both in learned representations and in the natural language processing literature (in the latter case commiting to particular functional forms with some free parameters). Moreover, the point of our Remark 2 is that some pre-commitment to a restricted composition function is essentially inevitable: if we allow the evaluation procedure to select an arbitrary composition function, the result will be trivial.

- Presentation of the approach: We are grateful for all the presentation suggestions. R2's summary of what they would like to be better stated is similar to the statement in the introduction that "the core of our proposal is to treat a set of primitive meaning representations D0 as hidden, and optimize over them to find an explicitly compositional model that approximates the true model as well as possible". We have updated the text of the paper to reinforce this point in several other places.

---

### Meta-Review · Area_Chair1 · 2018-12-14
**Worthwhile first step on an important problem, but clarity issues.**

**Confidence:** 3
**Recommendation:** Accept (Poster)

**Metareview:**

This paper presents a method for measuring the degree to which some representation for a composed object effectively represents the pieces from which it is composed. All three authors found this to be an important topic for study, and found the paper to be a limited but original and important step toward studying this topic. However, two reviewers expressed serious concerns about clarity, and were not fully satisfied with the revisions made so far. I'm recommending acceptance, but I ask the authors to further revise the paper (especially the introduction) to make sure it includes a blunt and straightforward presentation of the problem under study and the way TRE addresses it.

I'm also somewhat concerned at R2's mention of a potential confound in one experiment. The paper has been updated with what appears to be a fix, though, and R2 has not yet responded, so I'm presuming that this issue has been resolved.

I also ask the authors to release code shortly upon de-anonymization, as promised.